# Enterolith Treated with a Combination of Double-Balloon Endoscopy and Cola Dissolution Therapy

**DOI:** 10.3390/medicina59030573

**Published:** 2023-03-15

**Authors:** Kei Nomura, Tomoyoshi Shibuya, Masashi Omori, Rina Odakura, Kentaro Ito, Takafumi Maruyama, Mayuko Haraikawa, Keiichi Haga, Osamu Nomura, Hirofumi Fukushima, Takashi Murakami, Dai Ishikawa, Mariko Hojo, Akihito Nagahara

**Affiliations:** Department of Gastroenterology, Juntendo University School of Medicine, 2-1-1 Hongo, Bunkyo-ku, Tokyo 113-0033, Japan

**Keywords:** enterolith, double-balloon endoscopy, cola dissolution therapy

## Abstract

A 71-year-old woman with rheumatoid arthritis who had been taking NSAIDs for many years consulted our hospital for abdominal pain. She was diagnosed with a small bowel obstruction due to an enterolith according to an abdominal CT scan that showed dilation from the enterolith in the small intestine on the oral side. It was considered that the intestinal stone was formed due to stagnation of intestinal contents and had gradually increased in size, resulting in an intestinal obstruction. We performed antegrade double-balloon endoscopy (DBE) to observe and remove the enterolith. We used forceps and a snare to fracture the enterolith. During this attempt, we found a seed in the center of the enterolith. Since the intestinal stone was very hard, cola dissolution therapy was administered from an ileus tube for 1 week. The following week, DBE was performed again, and it was found that the stone had further softened, making attempts at fracture easier. Finally, the enterolith was almost completely fractured. Intestinal stenosis, probably due to ulcers caused by NSAIDs, was found. Small bowel obstruction with an enterolith is rare. In this case, it was considered that the seed could not pass through the stenotic region of the small intestine and the intestinal contents had gradually built up around it. It has been suggested that DBE may be a therapeutic option in cases of an enterolith. Further, cola dissolution therapy has been shown to be useful in treating an enterolith, with the possible explanation that cola undergoes an acid–base reaction with the enterolith. In summary, we report, for the first time, treatment of an enterolith with a combination of DBE and cola dissolution therapy, thereby avoiding surgery and its risks.

## 1. Introduction

Enteroliths are an uncommon medical condition [1], with prevalence ranging from 0.3% to 10% in selected populations. Various sized enteroliths are more common than anticipated because they typically remain underreported in the absence of clinical symptoms or due to their diminutive size that permits intermittent passage and may not always be visualized on radiologic images. The majority of enteroliths are discovered in symptomatic patients, who have abdominal pain or small bowel obstruction. Therefore, the prevalence of asymptomatic enteroliths is still largely unknown. Enteroliths are classified into primary and secondary types. Furthermore, primary enteroliths are divided into false and true enteroliths, with most classified as false. Primary false enteroliths have been shown to result from orally ingested substances, such as trichobezoar, phytobezoar, varnish stone, and fecalith. On the other hand, primary true enteroliths are originally created within the intestine by substances present, such as calcium and choleric acid. Conversely, secondary enteroliths occur from outside the intestine due to the migration of gall stones through a fistula. A detailed history and physical examination are required to diagnose enteroliths. Sudden or recurrent abdominal pain with vomiting in a patient who is in a population at risk for enteroliths should raise suspicion of the possibility of an enterolith. Important risk factors include intraluminal stricture or stenosis seen in Crohn’s disease, tuberculous and radiation enteritis; surgical anastomoses; intestinal malignancy; extraluminal kinking or angulation found in the setting of intra-abdominal adhesions, external compressions, or incarcerated hernias [1,2,3,4,5,6,7,8,9,10,11,12]. Radiological imaging has been useful for early diagnosis of enteroliths. Plain abdominal roentgenograms can detect stones in up to one-third of cases [13]. Computed tomography (CT) may also be useful in identifying the number of enteroliths and their exact location.

Optimal treatment of enteroliths should focus on enterolith removal and correction of the underlying pathology to prevent future formation of additional enteroliths. Enteroliths are asymptomatic in most cases. However, when symptoms appear, such as abdominal distension and/or abdominal pain, critical clinical conditions that require surgical treatment may arise because of ileus and intestinal perforation due to intestinal obstruction. Recently, it was suggested that double-balloon endoscopy (DBE) may be a therapeutic option in selected cases as it can approach the whole small intestine [14,15,16]. The benefit of using DBE for the treatment of enteroliths is that the risk of treatment complications is relatively lower than with surgery. Further, cola has been used to treat enteroliths [17,18]. Cola dissolution therapy has been useful for softening enteroliths and is simpler than surgery. Although each of these treatments is useful for treatment of enteroliths, the combination of DBE and cola dissolution therapy has not been reported. We, herein, present the first report of treatment of an enterolith using a combination of DBE and cola dissolution therapy.

## 2. Case Report

A 71-year-old woman with rheumatoid arthritis had been taking NSAIDs for many years. Seven years ago, she consulted our hospital for anemia. Gastroscopy and colonoscopy were normal and showed no evidence of bleeding. DBE was performed to detect small intestinal bleeding, and multiple small intestinal ulcers and membranous stenosis were found. Biopsy and stool cultures were negative; infections, vasculitis, and Crohn’s disease were ruled out. Moreover, since these findings had almost disappeared after not taking NSAIDs, she was diagnosed as having an NSAID ulcer in the small intestine. At the follow-up two years ago, a patency capsule was retained, and small bowel stenosis was found again by DBE. Aspirin, which had been started for coronary arteriosclerosis, was stopped, and balloon dilation was planned after the ulcer had healed. However, she did not return to our hospital for two years. Then, after two years, she consulted our hospital for abdominal pain from several days ago and was admitted. Blood testing revealed a leukocyte count of 5.3 × 10^9^/L and C-reactive protein level of 24.2 mg/L at admission. Abdominal X-ray revealed intestinal gas. As she had a history of an NSAID ulcer and stenosis in the small intestine, CT was performed to examine the intestine. She was diagnosed as having a small bowel obstruction by an enterolith according to an abdominal CT that showed dilation from the enterolith in the small intestine on the oral side (Figure 1a). It was considered that the intestinal stone was formed due to stagnation of intestinal contents and had gradually increased in size, resulting in intestinal obstruction. Upon the diagnosis of the small bowel obstruction, an ileus tube was placed and conservative treatment was administered. We recommended that she undergo surgery to remove the enterolith, but she refused surgery. Therefore, after intestinal decompression via the ileus tube and improvement in abdominal pain, we performed antegrade DBE (EN-580T^®^, Fujifilm, Tokyo, Japan) to observe and remove the enterolith (Figure 1b).

We used forceps (EndoJaw^®^, Olympus, Tokyo, Japan) and a snare (Snaremaster^®^, Olympus, Tokyo, Japan) to fracture the enterolith (Figure 2a,b). While attempting to fracture the enterolith, we found a seed at its center (Figure 2c). Since the intestinal stone was very hard, 500 mL/day of cola (Coca-Cola^®^) was injected from an ileus tube for 1 week to dissolve the stone. The following week, DBE was performed again, and the stone had further softened and was more easily fractured. Finally, the enterolith was almost completely fractured. Intestinal stenosis, probably due to ulcers caused by NSAIDs, was found (Figure 2d and Appendix A). After 1 week, we confirmed that there were no stones, and balloon dilation was performed. The patient began to eat and continued to be well upon discharge. Our follow-up of the patient has remained uneventful for three years.

## 3. Discussion

Small bowel obstruction by an enterolith is rare [1,19]. It is assumed that the formation of primary enteroliths is related to intestinal stenosis due to Crohn’s disease and/or intestinal tuberculosis. In addition, primary enteroliths may occur in the area of stasis due to the existence of intestinal diverticulum, afferent loops after surgery, incarcerated hernias, small intestinal tumors, and intestinal kinking from intra-abdominal adhesions [1,2,3,4,5,6,7,8,9,10,11,12]. In the present case, it was considered that the seed could not pass through the stenotic region of the small intestine due to the NSAID ulcer and the intestinal contents gradually built up around it. In most cases, surgical management is the main treatment for enteroliths because they are not discovered until the occurrence of clinical conditions, such as ileus or intestinal perforation. A comparatively large number of cases are diagnosed by laparotomy or autopsy [1]. Although it is difficult to diagnose asymptomatic enteroliths, once enteroliths are diagnosed, enteroscopy may become an effective but invasive treatment option. Several reports have suggested that DBE may be a therapeutic option in cases of an enterolith [15,16]. Moreover, DBE is useful in searching for an underlying pathology for enteroliths. In cases of intestinal stricture, stenosis, or an anastomotic defect, an attempt at endoscopic segment dilatation and stone retrieval, may be considered first [2,20]. Endoscopic snaring, electrohydraulic lithotripsy, and mechanical lithotripsy have been previously described [21,22,23,24]. It is generally believed that stones with a diameter > 25 mm may cause intestinal obstruction in the absence of luminal stricture or stenosis [25]. On the other hand, stones < 20 mm in diameter can pass through without symptoms. Thus, even if an enterolith cannot be removed completely, just fragmentizing it using a device through an enteroscope may be an effective treatment. However, it is possible that remnants may become a nidus for future stones. Importantly, enterolith formation may be the first clue to the existence of a compromised intestinal anatomy and every effort should be made to decrease future stone formation by recognizing and treating underlying medical conditions. Medical, endoscopic, or surgical correction of inflammatory, infectious, or structural pathology may provide chronic symptom relief and benefit the long-term outcome in many cases [26]. In the present case, balloon dilation was performed after fracturing the enterolith. Our follow-up of the patient has remained uneventful for a long time. The benefit of using DBE for the treatment of enteroliths is that the risk of treatment complications is relatively lower than surgery.

Further, cola dissolution therapy was reported as useful for an enterolith [17,18]. A possible explanation among its properties for dissolution is that cola undergoes an acid–base reaction with an enterolith. Ladas et al. reported that fine bubbles of carbon dioxide permeate the fine irregularities on the gastrolith surface and soften the fibrous bonds [27]. Cola dissolution therapy is a simpler treatment than surgery and is considered to be feasible in many facilities. However, the risks associated with injecting cola include mucosal damage due to carbonic acid and increased intestinal pressure due to carbon dioxide production. In particular, increased intestinal pressure during the acute phase of colitis may cause colonic perforation and bacterial translocation, leading to aggravation of sepsis. If intestinal necrosis occurs, prompt surgery is the principal treatment, and indications for surgery should be carefully considered.

In this case, it was difficult to fracture the enterolith using DBE because the intestinal stone was very hard. Generally, surgery is often selected for enteroliths in the small intestine if endoscopic treatment is not able to remove an enterolith. However, we ultimately administered cola to our patient via an ileus tube because she refused invasive treatment such as surgery. Cola dissolution therapy further softened the stone, making it easier to fracture, which may have served as an adjunct to endoscopic treatment and avoided surgery and its attendant risks. It is considered worth trying cola dissolution therapy before endoscopic treatment if enteroliths are hard and large.

## 4. Conclusions

In summary, we report, for the first time, treatment of an enterolith using a combination of DBE and cola dissolution therapy, thereby avoiding surgery and its attendant risks. This combination therapy may be a non-invasive treatment option for enteroliths.

## Figures and Tables

**Figure 1 medicina-59-00573-f001:**
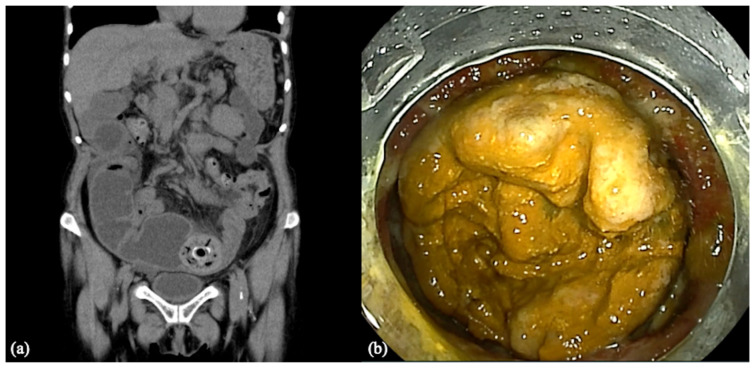
(**a**) Dilation of the small intestine and findings of an enterolith on CT. (**b**) Endoscopic view of the enterolith in the small intestine.

**Figure 2 medicina-59-00573-f002:**
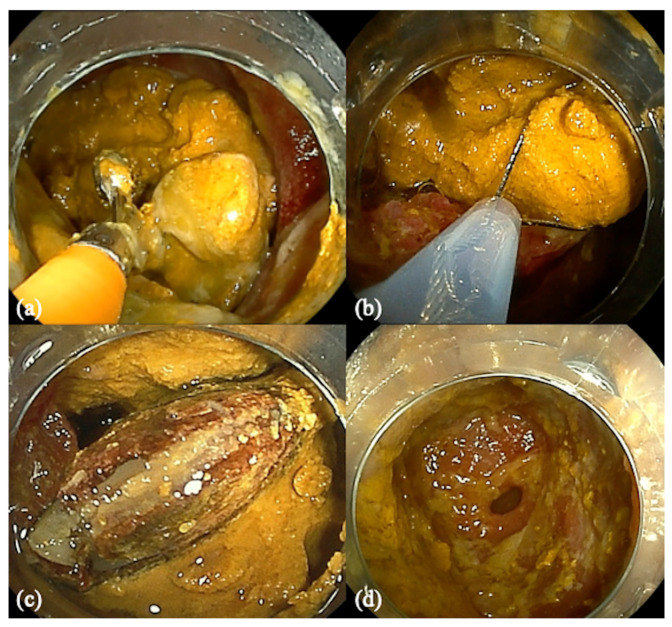
(**a**) We used forceps and (**b**) a snare in attempting to fracture the enterolith. (**c**) Seed in the center of the enterolith. (**d**) Small intestinal stenosis found after fracturing the enterolith.

## Data Availability

The data presented in this study are available on request from the corresponding author. The data are not publicly available due to patient’s privacy.

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
