# Peer review of "Enterolith Treated with a Combination of Double-Balloon Endoscopy and Cola Dissolution Therapy"

_medicina, 2023, doi:10.3390/medicina59030573_

Round 1

Reviewer 1 Report

The authors present the case of a 71-year-old female with an enterolith treated by a combination of double-balloon endoscopy and cola dissolution therapy.

The introduction is very poor and not informative. The authors should give more details in the introduction about pathogenesis of entheroliths. They should point out diagnostic dilemmas and include the differential diagnosis of this disease. Diagnostic and therapeutic modalities should be described in more detail.

The description of a case is also poor. Many important details are not presented, such as history (why the patient underwent CT, what symptoms occurred, how long the symptoms lasted...) and diagnostic procedures should be described in more details.

There is not even a word about the long-term follow-up of the patient.

The discussion is deficient and is largely a repetition of the existing literature. This is not a discussion point, it should be concise, compare this case to similar cases in the literature, and highlight key differences from previously published studies.

Finally this is just a simple case, nothing new, and similar pathology has been described several times in the literature. I see no benefit to readers in this report. The report of a simple case of enterolith is too premature.

Search of literature should be improved.

Author Response

Point 1: The introduction is very poor and not informative. The authors should give more details in the introduction about pathogenesis of enteroliths. They should point out diagnostic dilemmas and include the differential diagnosis of this disease. Diagnostic and therapeutic modalities should be described in more detail.

Response 1: We thank the reviewer for this constructive comment. The sentences, including pathogenesis, diagnosis and treatment of enteroliths, have been added in the introduction in order to describe in more detail.

Point 2: The description of a case is also poor. Many important details are not presented, such as history (why the patient underwent CT, what symptoms occurred, how long the symptoms lasted...) and diagnostic procedures should be described in more details.

There is not even a word about the long-term follow-up of the patient.

Response 2: We thank the reviewer for this comment. We have included many details of patient’s history. The sentences how to diagnose as NSAIDs ulcer have been added.

The following sentence about the long-term follow-up of the patient has been added in the case report.

‘‘Our follow-up of the patient has remained uneventful for three years’’.

Point 3: The discussion is deficient and is largely a repetition of the existing literature. This is not a discussion point, it should be concise, compare this case to similar cases in the literature, and highlight key differences from previously published studies.

Response 3: We thank the reviewer for this constructive comment. We have emphasized that cola dissolution therapy was suggested as a useful treatment before endoscopic treatment if enteroliths were hard and large, thereby avoided surgery and its attendant risks.

Point 4: Finally, this is just a simple case, nothing new, and similar pathology has been described several times in the literature. I see no benefit to readers in this report. The report of a simple case of enterolith is too premature.

Search of literature should be improved.

Response 4: We thank the reviewer for this comment. There have been no report of combination treatment of DBE and cola dissolution therapy. We have emphasized that cola dissolution therapy was suggested as a useful treatment before endoscopic treatment if enteroliths were hard and large, thereby avoided surgery and its attendant risks.

Reviewer 2 Report

The authors present a case report of an enterolith treated with non-surgical means. Enteroliths in human are a rare occurrence. As such, the scope of literature describing non-surgical treatment is limited, increasing the novelty and overall merit of the manuscript. The methodology is adequately described, the study design's is sound and conclusions drawn from the case report are concise and relevant.

Author Response

Point 1: The authors present a case report of an enterolith treated with non-surgical means. Enteroliths in human are a rare occurrence. As such, the scope of literature describing non-surgical treatment is limited, increasing the novelty and overall merit of the manuscript. The methodology is adequately described, the study design's is sound and conclusions drawn from the case report are concise and relevant.

Response 1: We thank the reviewer for this comment.

Reviewer 3 Report

The current case provide a small instestine stone treatment by enteroscopy and cola dissolution, which is relevant to clinic. But the diagnosis of stenosis by NSAID is not sufficient. Please provide more evidences on this.

Author Response

Point 1: The current case provides a small intestine stone treatment by enteroscopy and cola dissolution, which is relevant to clinic. But the diagnosis of stenosis by NSAID is not sufficient. Please provide more evidence on this.

Response 1: We thank the reviewer for this comment. The sentences which provide more evidence for diagnosis of stenosis by NSAIDs have been added in the case report.

Round 2

Reviewer 1 Report

The authors made some improvement to the case presentation. Discussion is still poor. Besides, as I pointed previously this is just a simple case, nothing new, and similar pathology has been described several times in the literature. I see no benefit to readers in this report.